# Enrichment of low-frequency functional variants revealed by whole-genome sequencing of multiple isolated European populations

Yali Xue[1,*], Massimo Mezzavilla[1,2,*], Marc Haber[1,*], Shane McCarthy[1,*], Yuan Chen[1], Vagheesh Narasimhan[1], Arthur Gilly[1], Qasim Ayub[1], Vincenza Colonna[1,3], Lorraine Southam[1,4], Christopher Finan[1], Andrea Massaia[1,5], Himanshu Chheda[6], Priit Palta[6,7], Graham Ritchie[1,8,9], Jennifer Asimit[1], George Dedoussis[10], Paolo Gasparini[11], Aarno Palotie[1,6,12,13,14,15,16], Samuli Ripatti[1,6,17], Nicole Soranzo[1,18], Daniela Toniolo[19], James F. Wilson[9,20], Richard Durbin[1], Chris Tyler-Smith[1] & Eleftheria Zeggini[1]

The genetic features of isolated populations can boost power in complex-trait association studies, and an in-depth understanding of how their genetic variation has been shaped by their demographic history can help leverage these advantageous characteristics. Here, we perform a comprehensive investigation using 3,059 newly generated low-depth whole-genome sequences from eight European isolates and two matched general populations, together with published data from the 1000 Genomes Project and UK10K. Sequencing data give deeper and richer insights into population demography and genetic characteristics than genotype-chip data, distinguishing related populations more effectively and allowing their functional variants to be studied more fully. We demonstrate relaxation of purifying selection in the isolates, leading to enrichment of rare and low-frequency functional variants, using novel statistics, $DVxy$ and $SVxy$. We also develop an isolation-index ($Isx$) that predicts the overall level of such key genetic characteristics and can thus help guide population choice in future complex-trait association studies.

[1] The Wellcome Trust Sanger Institute, Wellcome Genome Campus, Hinxton, Cambridgeshire CB10 1SA, UK. [2] Institute for Maternal and Child Health, IRCCS Burlo Garofolo, University of Trieste, 34137 Trieste, Italy. [3] Consiglio Nazionale delle Ricerche, Istituto di Genetica e Biofisica 'Adriano Buzzati-Traverso', via Pietro Castellino 111, 80131 Napoli, Italy. [4] Wellcome Trust Centre for Human Genetics, University of Oxford, Oxford OX3 7BN, UK. [5] National Heart and Lung Institute, Imperial College London, London SW7 2AZ, UK. [6] Institute for Molecular Medicine Finland (FIMM), University of Helsinki, Tukholmankatu 8, 00290 Helsinki, Finland. [7] Estonian Genome Center, University of Tartu, 23B Riia Street, 51010 Tartu, Estonia. [8] European Bioinformatics Institute, Wellcome Genome Campus, Hinxton, Cambridgeshire CB10 1SD, UK. [9] MRC Human Genetics Unit, MRC IGMM, University of Edinburgh, Western General Hospital, Crewe Road, Edinburgh EH4 2XU, UK. [10] Department of Nutrition and Dietetics, Harokopio University Athens, Athens, Eleftheriou Venizelou 70, Kallithea 176 76, Greece. [11] Medical Genetics, DSM, University of Trieste and IRCCS (Istituto di Ricovero e Cura a Carattere Scientifico) Burlo Garofolo Children Hospital, Via dell'Istria, 65, 34137 Trieste, Italy. [12] Analytic and Translational Genetics Unit, Department of Medicine, Massachusetts General Hospital, Boston, Massachusetts 02114, USA. [13] Program in Medical and Population Genetics, The Broad Institute of MIT and Harvard, Cambridge, Massachusetts 02114, USA. [14] The Stanley Center for Psychiatric Research, The Broad Institute of MIT and Harvard, Cambridge, Massachusetts 02114, USA. [15] Psychiatric & Neurodevelopmental Genetics Unit, Department of Psychiatry, Massachusetts General Hospital, Boston, Massachusetts 02114, USA. [16] Department of Neurology, Massachusetts General Hospital, Boston, Massachusetts 02114, USA. [17] Department of Public Health, University of Helsinki, Helsinki FI-00014, Finland. [18] Department of Haematology, University of Cambridge, Cambridge CB2 0XY, UK. [19] Division of Genetics and Cell Biology, San Raffaele Scientific Institute, via Olgettina 60, 20132 Milan, Italy. [20] Usher Institute of Population Health Sciences and Informatics, University of Edinburgh, Teviot Place, Edinburgh, EH8 9AG Scotland, UK. * These authors contributed equally to this work. Correspondence and requests for materials should be addressed to Y.X. (email: ylx@sanger.ac.uk) or to E.Z. (email: Eleftheria@sanger.ac.uk).

Population variation in disease susceptibility has been shaped by environment, demography and evolutionary history. Isolated populations (isolates) have generally experienced bottlenecks and strong genetic drift, so by chance some deleterious rare variants have increased in frequency while some neutral rare variation is lost, both helpful characteristics for the discovery of novel rare variant signals underpinning complex traits[1–3]. Studies to date have focused on individual isolates and have identified several disease-associated signals[4–12]. However, isolates differ in the time when they became isolated, their initial population size, the level of gene flow from outside and other historical demographic factors, and consequently also differ in their power for association studies[2]. We thus generate and analyse low-depth (4 × –10 ×) whole-genome sequences (WGS) from eight cohorts drawn from isolated European populations and compare each isolate with the closest non-isolated (general) population, for which we also generate or access WGS data. We then investigate empirically how these historical differences influence the population-genetic properties of isolates, and frame these insights in terms of their consequences for study design in complex trait association studies.

## Results

**Samples, sequencing and QC.** The data set includes newly generated low-depth (4x–10x) WGS from eight cohorts drawn from isolated European populations: one each from Kuusamo in Finland (FIK) and Crete in Greece (GRM[13]), four from Friuli-Venezia Giulia in Italy (IF1, IF2, IF3 and IF4 (ref. 14)), and one each from Val Borbera in Italy (IVB[15]) and the Orkney Islands in the UK (UKO[16]); and the closest non-isolated (general) population: Finland (FIG[9]), Greece (GRG), together with publicly available data for Italy (ITG[17]) and UK (UKG[18]) (Fig. 1a and Supplementary Table 1). We generated a superset of variants called in these cohorts and all 26 population samples in the 1000 Genomes Project Phase 3 (ref. 17), and performed multi-sample genotype calling across all 9,375 samples (3,059 from the current study, 2,353 from the 1000 Genomes Project Phase 3 release and 3,781 from UK10K). Both individual population and amalgamated genotype call data, which have greater than 99% concordance with genotyping data (Supplementary Table 2), are available to the scientific community (Data availability).

**General description of the variants in the isolates.** We identified approximately 12.2 million variants with minor allele frequency (MAF) ≤2% (rare), 5.5 million with MAF >2–≤5% (low-frequency) and 8.3 million variants with MAF >5% (common) across the ten populations newly sequenced here (eight isolates, GRG and FIG). Of these, 10.5, 0.7 and 0.3%, respectively, are novel (Table 1 and Supplementary Table 3). As expected, most of the isolates have lower numbers of variant sites per genome than their closest general population (Supplementary Fig. 1, Supplementary Table 5). We find ~188,000–~513,000 variants that are common with MAF >5.6% in each isolate but with MAF ≤1.4% in its closest general population (Table 1); ~30,000–122,000 of these per isolate have frequency ≤1.4% in all the general samples studied, among which ~150–~700 in coding regions and ~500–~2,800 genome-wide are deleterious (Supplementary Table 4). These common and low-frequency variants are thus useful markers for whole-genome association studies in these populations and some of them (if absent from the general population) could potentially lead to novel association signals. They include known examples such as rs76353203 (R19X) in APOC3 in GRM, which is associated with high-density lipoprotein and triglyceride levels[6].

**Population-genetic analyses in the isolates.** Previous population-genetic studies of isolates have, with some exceptions[11,19], been based on common variants found on genotyping arrays, and have illustrated general characteristics such as low genetic diversity and longer shared haplotypes[9,13–15,19,20]. Rare variants discovered from sequencing are on average more recent in origin than common variants[21] and therefore more powerful for distinguishing closely related populations and more informative about recent demographic history. We find that isolates are, as expected, genetically close to their matched general population in principal component analyses (PCA), ADMIXTURE[22] and TreeMix[23] using common variants (Fig. 1b, Supplementary Figs 2–5 and Supplementary Table 6), but PCA using rare and low-frequency variants, as found previously[24], distinguishes them more clearly from the general population and also from other isolates, particularly among the Italian samples (Fig. 1c, Supplementary Fig. 2). The majority of sharing of variants present just twice across all samples of 36 individuals from each population ($f_2$ variants[21]) takes place within the same population, and the isolates generally share more with their closest general population than with other populations. This latter trend, however, is not apparent for IF1–IF4, who show little sharing with any other population, pointing to a greater level of isolation and lower level of gene flow with their general population (Fig. 1d, upper triangle and Supplementary Fig. 7), which is confirmed by f3-statistics[25] comparing with a worldwide population panel of HGDP-CEPH samples using common SNPs (Supplementary Fig. 6). $f_3$–$f_{10}$ variant sharing demonstrates sharing by ITG and IVB with both Greek and UK populations (Fig. 1d, lower triangle and Supplementary Fig. 7), potentially indicative of their more ancient heritage.

**Population demographic history.** All populations studied here, both isolates and general, appear to have shared a comparable effective population size (Ne) history before 20 thousand years ago (KYA) based on the multiple sequentially Markovian coalescent method[26] (Supplementary Fig. 9). The isolates diverged from their general populations within the last ~5,000 years based on LD estimations[27] (Supplementary Table 7 and Supplementary Fig. 8) and yet had sharp decreases in their population sizes in more recent times as estimated using inferred long segments of identity by descent (IBD)[28] (Fig. 1e,f and Supplementary Fig. 10). Different isolates also split from their respective general populations at different times. For example, IF1–IF4 split from ITG ~4–5 KYA, while most other isolates split from their general populations within the last ~1,000 years (Supplementary Table 7).

The different demographic histories of different isolates should lead to different genetic characteristics. To summarize these features in a single quantitative measure that can be calculated from genotype data, as well as sequence data, we developed an isolation index (Isx) which combines information on the divergence time from the general population (Tdg), Ne and migration rate (M), such that early-divergence-time isolates with small Ne and low M have a high Isx value (Fig. 2a and Supplementary Fig. 11). The different isolates show different Isx values: IF1, IF2, IF3 and IF4 have the highest, while IVB has the lowest (Supplementary Table 8). Isx values are highly correlated with other population-genetic characteristics (for example, Fig. 2b,c, Supplementary Table 11), such as genome-wide pairwise $F_{ST}$ between isolates and their matching general population (reflecting the genetic drift of the isolates) (Supplementary Fig. 12), the total length and number of runs of homozygosity (ROH) (Supplementary Fig. 13), inbreeding coefficient (F) (Supplementary Fig. 14) and length of LD

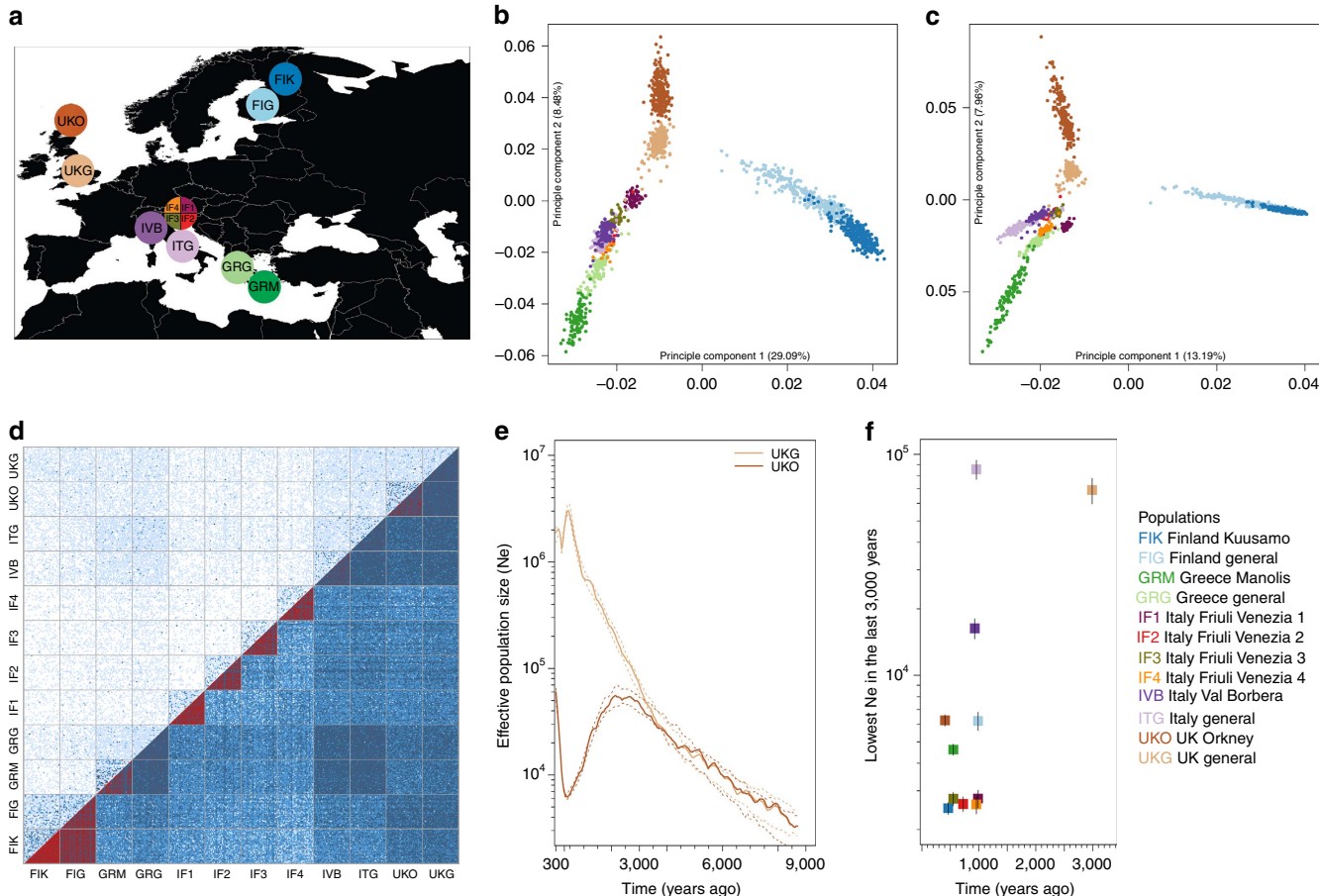

**Figure 1 | General characteristics and demographic history of isolated and matched general populations. (a)** Geographical locations of samples. The base map was plotted in R using the mapdata package and circles were added using Photoshop. **(b)** PCA using common variants. **(c)** PCA using low-frequency variants. **(d)** Sharing of rare variants within and between populations. Upper left triangle: $f_2$ variants; lower right triangle $f_3$–$f_{10}$ variants. **(e)** Effective population size (Ne) inferred from IBDNe for UKO and UKG during the past nine KY. **(f)** The lowest Ne inferred by IBDNe for all populations for the past three KY, plotted as a function of the time at which it occurred.

**Table 1 | Summary of variants discovered in this study.**

| POP | n | average depth | MAF ≤ 2% | | MAF > 2–≤ 5% | | MAF > 5% | | Novel common SNPs in isolate* | Novel common SNPs in isolate† |
|---|---|---|---|---|---|---|---|---|---|---|
| | | | total | novel% | total | novel% | total | novel% | | |
| FIK | 377 | 4x | 4,066,373 | 10.90 | 1,553,076 | 1.20 | 6,025,077 | 0.70 | 190,527 | 70,579 |
| FIG | 1,564 | 6x | 6,548,833 | 11.80 | 1,540,915 | 0.80 | 6,053,704 | 0.70 | na | na |
| GRM | 249 | 4x | 5,129,513 | 7.20 | 1,447,981 | 1.10 | 6,111,923 | 0.80 | 513,272 | 49,884 |
| GRG‡ | 99 | 10–30x | 3,757,110 | na | 1,321,955 | na | 5,842,537 | na | na | na |
| IF1 | 60 | 4–10x | 1,456,881 | 1.30 | 1,420,929 | 1.30 | 5,890,714 | 0.80 | 320,191 | 119,157 |
| IF2 | 45 | 4–10x | 1,063,098 | 1.30 | 1,554,145 | 1.00 | 6,001,568 | 0.80 | 273,694 | 94,496 |
| IF3 | 47 | 4–10x | 961,059 | 1.30 | 1,455,284 | 1.10 | 6,068,304 | 0.80 | 299,603 | 107,281 |
| IF4 | 36 | 4–10x | 1,030,673 | 1.30 | 1,124,789 | 1.10 | 6,001,625 | 0.80 | 308,356 | 122,254 |
| IVB | 222 | 6x | 4,857,767 | 1.60 | 1,396,799 | 0.80 | 6,112,476 | 0.80 | 188,972 | 30,284 |
| UKO | 397 | 4x | 5,963,416 | 11.70 | 1,471,782 | 0.80 | 6,047,383 | 0.80 | 193,300 | 36,512 |
| Total | 3,096 | | 12,218,797 | 10.50 | 5,503,179 | 0.70 | 8,301,524 | 0.30 | | |

'Novel' variants are those not found in 1000 Genomes Project Phase 3 or UK10K project.
*Variants that are common (minor allele frequency, MAF ≥ 5.6%, alternative allele count ≥ 4) in an isolated population but not common (MAF < 1.4%, alternative allele count ≤ 1) in its closest general population.
†Variants that are common (MAF ≥ 5.6%, alternative allele count ≥ 4) in an isolated population but not (MAF < 1.4%, alternative allele count ≤ 1) in any of the general populations.
‡Different variant calling procedure in this population.

(Supplementary Figs 15 and 16 and Supplementary Tables 9 and 10). All these characteristics are correlated, but the pairwise correlation coefficients show that *Isx* is a slightly better overall predictor of the other measures than any single existing measure

(Fig. 2c, Supplementary Fig. 17 and Supplementary Table 11); moreover, it is potentially more robust to confounding factors as it is calculated from three demographic parameters, while the others are all based on single measurements.

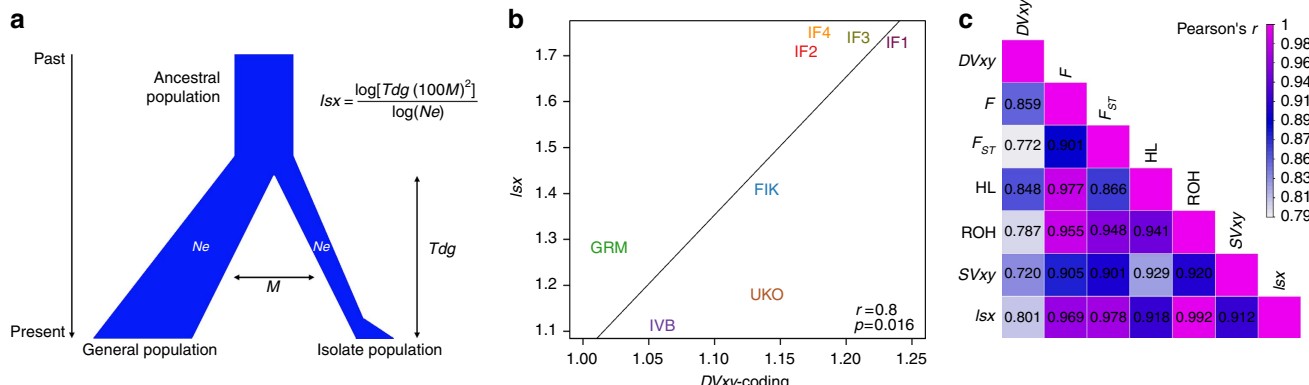

**Figure 2 | Isolation index (*Isx*) and its correlation with other genetic measures.** (**a**) Information summarized in *Isx*. (**b**) Example of the correlation between *Isx* and other statistics, here *DVxy-coding*. (**c**) Summary of the correlations between *Isx* and other population-genetic statistics. All the correlation coefficients are high and statistically significant.

**Purifying selection analyses.** Several lines of evidence suggest relaxed purifying selection in the isolates due to their reduced *Ne*, although as expected we do not detect substantially increased genetic load per genome using the *Rxy* statistic[29] based on all of the variants in the genomes (Fig. 3a and Supplementary Table 12). First, we see different levels of enrichment of low-frequency functional variants in isolates (Fig. 3b,c, Supplementary Tables 13 and 14, Supplementary Fig. 18a) quantified by a new statistic, *DVxy-coding*, developed here (DV: drifted variants). *DVxy-coding* measures the ratio of functional coding variants (missense plus loss-of-function (LoF)) in isolates compared to the closest general population (and vice-versa), adjusted for the corresponding ratios of intergenic variants in order to correct for the effect of genetic drift. We applied this only to a subclass of DVs, defined as low-frequency (2–5%, the best choice according to the sample size we have) in any isolate, yet at least three-fold higher than in the closest general population (and vice versa). We find that *DVxy-coding* is >1 in all isolates and <1 in all general populations (Fig. 3c, Supplementary Fig. 18a and Supplementary Table 13). We also calculated a similar *DVxy-wg* statistic by stratifying whole-genome variants according to their combined annotation dependent depletion (CADD) score (0–5, neutral variants; 5–10, mildly deleterious; 10–20, deleterious; and >20, highly deleterious; these cut-off choices balance the number of variants in each bin to allow us comparable statistical power among all bins, although the conclusions are robust to the particular cut-off values chosen and different bins (Supplementary Figs 18b and 19)). The *DVxy-wg* values are differentiated for variants with CADD score of 10–20 and significantly so (assessed using the jack-knife bootstrap method) for ones with CADD scores >20, with *DVxy-wg* values >1 in all isolates and <1 in all general populations (Fig. 3b, Supplementary Fig. 18b and Supplementary Table 14). This demonstrates enrichment of low-frequency functional variants, both coding and genome-wide with CADD score >10, in the isolated populations. Moreover, both *DVxy-coding* and *DVxy-wg* values are correlated with *Isx*, suggesting that different isolation characteristics lead to different levels of enrichment of functional variants.

We also investigated the relaxation of purifying selection by assessing functional (missense) singleton variants (SV) pooled for all of the genes that have at least one singleton missense or synonymous variant in a pair of populations (one isolate and its general population), correcting with pooled synonymous variants (*SVxy* statistic,). We find a substantial deviation from 1 for functional singletons in all of the isolates (Fig. 3d and

Supplementary Table 15), with *SVxy* values positively correlating with *Isx* (Fig. 2c and Supplementary Fig. 20). We also find that the proportion of relaxed essential genes[30] with *SVxy* >1 in isolates is significantly higher than in the general population (Supplementary Table 15). Such rare and low-frequency drifted functional variants, measured by both *SVxy* and *DVxy*, are particularly relevant for boosting the power of association studies[6].

**Positive selection analyses.** We do not find convincing evidence for positive selection in any isolate using deltaDAF[31], PCAdapt[32] or singleton density score (SDS)[33], although we do identify some highly differentiated variants (Supplementary Fig. 21 and Supplementary Tables 16 and 17), including in the protein-coding genes *ALK*, *SPNS2*, *SLC39A11* and *ACSS2*, which can nevertheless be accounted for by drift. Interestingly, we also find six highly differentiated variants shared between different isolates from Italy, IF2, IF3 and IF4, but interpret them as likely to result from drift or positive selection for the ancestral allele in the ITG (Supplementary Table 17). We find that the SDS method has little power in our samples because of their small size, and failed to detect selection even at the lactose tolerance SNP in the UKO, a known strong signal of recent selection (Supplementary Fig. 22).

**Discussion**
Isolated populations have special characteristics that can be leveraged to increase the power of association studies, as several previous studies have shown[19,34]. Nevertheless, only a small proportion of functional variants have increased in frequency in any one isolate, so multiple isolates must be investigated to reveal the full diversity of associated variants. Here, we probed an extended allele frequency spectrum of variants potentially underpinning human complex disease through the analysis of WGS data in multiple isolates matched to nearby non-isolated populations, capturing common, low-frequency and rare variants. We quantified different levels of isolation resulting from different demographic histories and have demonstrated that the *Isx* statistic, calculated even from SNP-chip data, reliably captures these relevant features. This study provides a systematic evaluation of the genetic characteristics of multiple European isolates and for the first time empirically demonstrates enrichment of rare functional variants across multiple isolates. With the advent of large-scale whole-genome sequencing, studies in isolates are poised to continue as major contributors to our understanding of complex disease etiology.

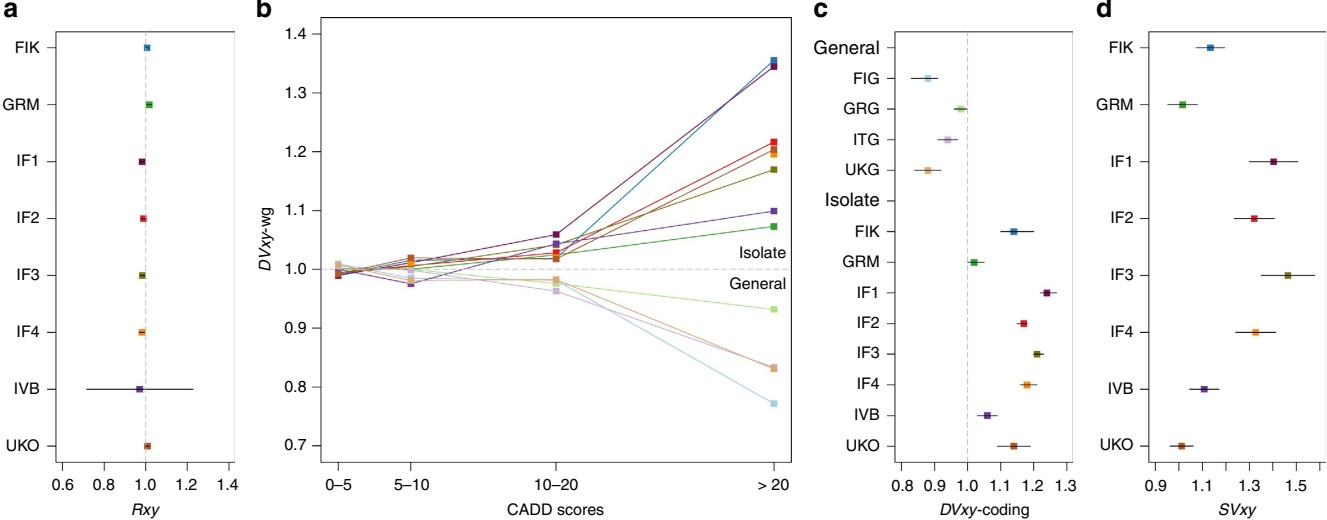

**Figure 3 | Purifying selection in the isolates and general populations. (a)** Rxy-missense statistic in each isolate, showing no evidence for increased genetic load in the isolates. The mean and s.d. for each Rxy value from 100 bootstraps are shown. **(b)** DVxy-wg (DVxy-whole genome) statistic in isolates and general populations, stratified by CADD score, showing enrichment of highly functional low-frequency variants. **(c)** DVxy-coding statistic in isolates and general populations, showing enrichment of low-frequency missense variants in isolates. **(d)** SVxy-missense statistic in each isolate, showing relaxation of purifying selection in isolates in singletons. The s.e.'s for both DVxy and SVxy were calculated by randomly sampling data from 20 chromosomes 100 times. All of these analyses are based on the minimum-sample-size data set (36 individuals from each population).

## Methods

**Data set and variant calling.** The data set includes 3,059 whole-genome low-depth sequences generated at The Wellcome Trust Sanger Institute using the Illumina Genome Analyzer II and Illumina HiSeq 2000 platforms, as well as 100 high-depth sequences from the Illumina HiSeq X Ten (Fig. 1a and Supplementary Table 1). Informed consent was obtained from all subjects and the study was approved by the HMDMC (Human Materials and Data Management Committee) of the Welcome Trust Sanger Institute. The multi-sample genotype calling across all of the low-coverage sequencing data from the current study, as well as 2,353 from the 1000 Genomes Project Phase 3 release, and 3,781 from UK10K (a total of 9,375) was performed with the defined site selection criteria (Supplementary Note). Genotype likelihoods were calculated with samtools/bcftools (0.2.0-rc9) and then genotypes were called and phased using Beagle v4 (r1274) (ref. 35). We assessed the performance of the genotype calling from the low coverage data using the available genotype chip data for a subset of the cohorts consisting of 4,665 individuals, and calculated the discordance rates on chromosome 20 separately for the categories REF-REF, REF-ALT and ALT-ALT.

The sample sizes are very different across these collections, and we used three different standard-sized subsets of the samples for different analyses: (1) the whole data set; (2) the sample-size-matched data set, obtained either by randomly selecting samples from general population to match the isolated population (for example, we randomly select 377 from FIG to match FIK), or by randomly selecting a subset of the isolated population to match the general population (for example, we randomly select 108 IVB to match the general population ITG); (3) the minimum-sample-size data set of 36 individuals per population. By doing this, we maximize the use of the data for different analyses, and we specify which data set is used for each analysis. The sequencing depth is also different across different populations, within a 2.5-fold range (apart from GRG, in which variants were called differently, details in Supplementary Notes), and we allowed for these differences when interpreting the results.

**Variant counts.** We first re-annotated all variants using the Variant Effect Predictor annotation from Ensembl 76 with the '- pick' option, which gives one annotation per variant. We then performed variant counting at both the population and individual level, stratifying by functional categories and frequency bins. These counts were either plotted in figures or summarized as median values in tables. We carried out these analyses using both the sample-size-matched data set and the minimum-sample-size data set.

**Population-genetic analyses.** We used the whole data set for the analyses in this section, unless otherwise specified. PCAs were performed separately with common variants or rare variants using EIGENSTRAT v.501 (ref. 36). Shared ancestry between the populations studied here was evaluated using ADMIXTURE v1.22 (ref. 22). The relationships between the populations studied here, combined with worldwide populations from the HGDP-CEPH panel[37], were also examined using

ancestry graph analyses implemented in TreeMix v.1.12 (ref. 23). We also used formal test of f3-statistics[25] to investigate population mixture in the history of the populations studied here, as well as worldwide populations from the HGDP-CEPH panel. Rare $f_2$ variants (with only two copies of the alternative allele in the minimum-sample-size data set) and moderately rare $f_{3-10}$ variants (3–10 copies of the alternative allele in the same data set) are particularly informative for investigating recent human history[21]. We investigated the sharing pattern of these two types of variant by summing all $f_2$ variants or any random two alleles of the $f_{3-10}$ variants shared by pairs of individuals. We plotted the results as a heat map using the image[1] function from the base R package (https://stat.ethz.ch/R-manual/R-devel/library/graphics/html/image.html). Variants were aggregated by pair of individuals using the 'count' function of the plyr package, then arranged in matrix form and colourized using 'colorRampPalette' from the colorspace package (https://cran.r-project.org/web/packages/colorspace/index.html). ROH, inbreeding coefficient (F) as well as the length of LD-blocks were calculated in PLINK, and finally genome-wide $F_{ST}$ values between isolates and their general populations were calculated with the software 4P (ref. 38) using the minimum-sample-size data set.

**Demographic inferences.** LD-based[39–41] demographic inference was performed in the NeON R package[27] using the minimum-sample-size data set; the median and confidence interval were estimated using the 50th, 5th and 95th percentiles of the distribution of long-term Ne in each time interval. We used the multiple sequentially Markovian coalescent method[26] to infer demographic changes before 20,000 years ago using four individual sequences from each population. In order to account for some loss of heterozygous sites in the low-depth data, we used a slow mutation rate of $0.8 \times 10^{-8}$ mutations per nucleotide per generation and a longer generation time of 33 years. We then estimated more recent demographic changes (from the present to ~9,000 years ago) using IBDNe[28] with the minimum-sample-size data set. We used IBDseq[42] to detect IBD segments in sequence data from chromosome 2 in all populations. We then used IBDNe with the default parameters and a minimum IBD segment length of 2 centiMorgan (cM) units. We assumed a generation time of 29 years.

**Isolation index.** In order to quantify the different isolation levels of different isolates, we developed an index that combines three demographic parameters: (a) Tdg, (b) Ne and (c) the level of private isolate ancestry (M). We call this estimate the Isolation index (Isx). It is defined as:

$$Isx = \frac{\log\left(Tdg(100 \times M)^2\right)}{\log(Ne)}$$

Both Tdg and Ne were inferred from the LD-based method using the NeON R package[27]. M is difficult to estimate directly from SNP genotype data, so here we estimated the difference of shared ancestral components between an isolate and its general population from ADMIXTURE analysis. We ran ADMIXTURE with only one isolate and it closest general population using K = 2. We then estimated the

difference in the means of ancestry between the isolate and its general population. The *M* parameter was defined as Delta Ancestry.

**Rxy analysis.** *Rxy* statistics[29] between each pair of populations (an isolate and its closest general population) for different functional categories were calculated using the matched-sample-size data for missense and LoF variants, including stop gain, splice donor and acceptor variants, using synonymous variants as controls (we did not use intragenic variants as control because of the ascertainment in the ITG which has high-depth exome sequences and low depth for the rest of the genome). We also calculated *Rxy* statistics for variants with CADD scores[43] greater than 10 and 20, using variants with CADD scores less than 5 as controls. The mean and s.d. for each *Rxy* value were obtained from 100 bootstraps.

**DVxy analysis.** A new statistic, *DVxy*, was developed to quantify the enrichment of low-frequency functional variants in the isolates using both the matched-sample-size and minimum-sample-size data sets. It calculates the proportion of functional variants in each isolate compared with its general population, correcting for genetic drift at the same time. We calculated *DVxy* specifically for the subset of variants with DAF 2–5% in the isolate, and at least three times lower in its closest general population, or vice-versa. We called these variants 'drifted variants' (DV). *DVxy* was calculated for both coding regions and whole genomes.

For coding variants, we defined missense or missense plus LoF variants as functional variants. We counted the number of functional DVs and neutral (intergenic) DVs in each isolate (population *x*) and the corresponding general population (population *y*). The ratio between the fraction of DV variants from the isolated population (corrected by the count of intergenic variants) and the corresponding fraction of DV variants from its general population was defined as the *DVxy* statistic. If *DVxy* is equal to 1, there is no enrichment for the functional DVs in the isolate; less than 1 indicates depletion, and greater than 1 indicates enrichment.

$$DVxy\_coding = \frac{\% \, DVx \, missense}{\% \, DVx \, intergenic} \Big/ \frac{\% \, DVy \, missense}{\% \, DVy \, intergenic}$$

For the whole genome, we used different CADD score cut-offs and bins. We calculated a DV statistic by stratifying the variants according to their CADD scores (0–5, neutral variants; 5–10, mildly deleterious; 10–20, deleterious; and greater than 20, highly deleterious) for each isolate and its closest general population. We finally calculated a ratio of the fraction of DV variants (from each class) between the isolate and its general population, and vice-versa. The following formula shows the *DVxy-wg* calculation for variants with CADD score between *i* and *j* in an isolate and its general population.

$$DVxy_{CADD(ij)} = \frac{\% \, DVx(CADD \, i - j)}{\% \, DVy(CADD \, i - j)}$$

The 95% confidence interval for each calculation was obtained by randomly sampling data from 20 chromosomes 100 times.

**SVxy analysis.** We further investigated the relaxation of purifying selection in the isolated populations using SVs. Here, we also used the minimum-sample-size data set. Another new statistic, *SVxy*, was developed to measure the ratio of missense versus synonymous singletons per gene in each population, as well as the ratio of the sum of singletons in all genes which have at least one singleton in the pair of the populations (one isolate and one general population). We counted the number of missense singletons and synonymous singletons per gene in each population, and *SVgene* was calculated as:

$$SVgene = \frac{(SV \, missense \, count + 1)}{(SV \, synonymous \, count + 1)}$$

*SVgene* > 1 indicates relaxation of purifying selection; *SVgene* = 1 indicates neutrality; and *SVgene* < 1 indicates purifying selection.

We then divided the gene list into essential genes[30] and non-essential genes (the rest), and calculated a statistic, *G_SV*, for each population, defined as:

*G_SV* = percentage of essential genes with *SVgene* > 1/percentage of non-essential genes with *SVgene* > 1

We finally calculated a statistic, *SVxy*, which is the ratio of *SVpop* of each isolate to *SVpop* of its general population. *SVpop* for each isolate and its general population was calculated using all genes which have at least one singleton in the pair of the populations and defined as *SVpop* = Σ (SV missense counts)/Σ(SV synonymous counts).

We used the same annotation as in the variant counts. We calculated a confidence interval for each estimate using bootstrapping of 80% of the genes 100 times.

**Correlation analyses.** We calculated pair-wise correlation coefficients between the *Isx* values, population-genetic measurements ROH, F, *F_ST*, and number and length of LD blocks, as well as the newly developed statistics *DVxy* and *SVxy* using the Pearson correlation in R.

**Positive selection analyses.** We calculated genome-wide pairwise derived allele frequency differences (deltaDAF) for each pair of populations (an isolate and its general population) as described previously[31] using the matched-sample-size data set. We also carried out PCAdapt analyses[32] for each pair of populations using the whole data set. Both analyses look for high derived allele frequency variants in the isolates, and will not be affected by sample size. Finally, we ran the SDS method[33] using the whole UKO and UKG data sets, which have the largest sample sizes for both isolate and its general population, and thus the greatest power for this method.

**Data availability.** Amalgamated genotype calls across all populations studied are available through the European Genome/Phenome Archive (EGAD00001002014) with Data Access Agreement described in Supplementary Information.

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

## Acknowledgements

We thank all study participants for making this work possible. Our work was supported by the Wellcome Trust (098051). We also thank the UK10K Consortium and SISu Consortium for making their data available to this study; detailed acknowledgements for these contributions are included in Supplementary Information.

## Author contributions

Y.X., C.T.-S., R.D. and E.Z.: design and supervision of the project. G.D., P.G., A.P., S.R., N.S., D.T. and J.F.W.: population liaison, sampling and DNA provision. N.S., J.F.W. and R.D.: comments and approval of the manuscript on behalf of the population consortia. Y.X., M.M. and M.H.: statistical method development. M.M., M.H., S.M., V.N., A.G., Q.A., V.C., L.S., C.F., G.R., H.C. and P.P. and J.A.: population-genetic analyses, statistical analyses and data interpretation. Y.C. and A.M.: bioinformatics support. S.M., N.S. and R.D.: data processing and QC. Y.X., M.M., M.H., S.M., V.C., C.T-.S. and E.Z.: manuscript drafting. All authors: approval of the final version of the manuscript.

## Additional information

**Competing interests:** The authors declare no competing financial interests.

**Publisher's note**: 

 7