## [Peer Review File · Nature Communications]

Reviewers' comments:

Reviewer #1 (Remarks to the Author):

In their study, Xue et al. analyzed germline variants in 10 isolated populations in comparison with variants in the general population from the 1000 Genomes and UK10K projects. Well executed and described in great detail, albeit mostly in the supplementary material, this large study can serve as a highly useful resource that will no doubt be applied in many future studies.

At the same time, the results presented are incremental, which poses a major weakness of providing too few biological insights. The authors introduce two new measures— DV_{xy} and SV_{xy} —to quantify the diminishing purifying selection in isolates, as well as an additional measure for isolatedness (i.e., Isx), which they suggest can “guide population choice in future complex trait association studies.” Although a potentially important finding, the authors failed to demonstrate how these measures can guide population selection. In response, can the authors provide an example of how using Isx can dramatically improve association studies that use isolated populations instead of the general population?

Minor comments

1. It is unclear to me which samples were included in the total count of 9,375 from the current study and the 1000G and UK10K projects.
2. Although I might have overlooked such an explanation, it appears as though the term N_e is not defined in the text.

Reviewer #2 (Remarks to the Author):

This is an interesting study of whole genome sequencing of multiple isolated European populations along with matched general population controls. The idea was to perform low-coverage sequencing (4-10X) of ~3000 individuals representing a variety of populations, reconstruct their demographic history, and quantify the extent of population-specificity of variants in each isolated population. The authors develop several new statistics to describe the isolatedness and population specificity of variants to discuss the potential relaxation of selective constraint on population isolates. Importantly, the data are made available through EGA. I very much like seeing the demographic inference of these populations, and think the data are summarized well to have broad impact. Of course there are usually several ways that the study could be improved, and I've tried to identify them below.

First, I don't think the title of the manuscript reflects the results of the paper. The only aspect of “functional” variants in this paper appears to be the use of CADD scores. This is of course a functional prediction, not a true aspect of “function”. Further, it is not clear why CADD score >20 is used as a threshold. Lastly, it is not clear that there are any tests of “enrichment” statistically backing up the title's claim.

Second, I would argue that the authors should be more precise with their use of terminology when discussing relaxation of purifying selection. One measure of natural selection is the fitness effect of a given mutation (s). This is not something that is observable, and is actually quite hard to infer. The “strength” of selection is often thought of in terms of the product of the fitness effect and the effective population size (N_e*s). For a population isolate, the extreme bottleneck and subsequent inbreeding can result in a sustained reduction in N_e , which of course reduces the product of N_e*s , but such a bottleneck or any other demographic effect is unlikely to impact s . The authors find that, consistent

with Do et al [29], there is no evidence for reduction of genetic load across individuals. It therefore seems incorrect to conclude (as per the abstract) that the authors “demonstrate the relaxation of purifying selection”. In fact, what the authors are seeing is that some deleterious variants increase in frequency due to drift, but the small population size results in a reduction in the introduction of new deleterious alleles (thereby having no effect on genetic load). This is a result that is also similar to Simons et al. [PMC3953611].

Major comments:

On page 4, the authors argue that IF1-IF4 exhibit a “greater level of isolation and lower level of gene flow with their general population”. However, IF1-IF4 have the smallest sample size (by a factor of >6 on average). I am worried that comparisons like this and other places that differences in sample size could be playing a major role in population differences. The general populations that the authors sample are far larger than any of the isolates, and so it is unclear to what extent the results are driven by sample size. For some of the analyses, I think it would be very helpful to have a supplemental figure showing what happens to figures 1d, 2b, 3c, and 3d if all populations were down sampled to 36 individuals (the size of IF4, your smallest sample size in Table 1).

Around lines 166, the authors say “several lines of evidence suggest relaxed purifying selection in the isolates due to their reduced N_e , although as expected we do not detect substantially increased genetic load per genome...”. It seems that these are directly contradictory statements. I don’t see how you can have relaxed purifying selection without increased load. Instead, it seems like you have equivalent levels of purifying selection, but increased drift due to lower N_e . It would be great to see how the authors differentiate relaxed purifying selection from a lower N_e that causes increased drift (so most deleterious variants are lost, but some increase in frequency) and a decreased introduction of novel deleterious variants. The statistics proposed do not seem distinguish these two, and seem consistent with my alternative explanation. I would argue that your results are highly interesting and valuable either way, but one just needs to be careful with terminology to be consistent with recent literature unless you have strong evidence refuting recent literature (and I do not see that being the case here).

The authors suggest that they do not find convincing evidence for positive selection in any isolate using two statistics. While this is consistent with some arguments in the literature (e.g. Hernandez et al [PMC3669691]), it would also be helpful to try a couple other statistics, particularly the SDS metric from Field et al [PMID: 27738015] recently presented (likely published after submission of this paper, though the preprint and method have been available for a while).

Minor comments:

I am not a fan of approximate numbers in an abstract, and think people would appreciate seeing “Here, we perform a comprehensive investigation using low-depth whole genome sequencing of 1433 individuals from eight European isolates and 1663 individuals from two matched general populations.” You then integrate data from a couple other population isolates and 1000 Genomes Project data, and I’m not sure how best to indicate that here.

In the first paragraph, the authors say “In isolated populations (isolates), some deleterious rare variants have increased in frequency while some neutral rare variation is lost...”. The first part is likely to be a minority of cases, but I don’t see this quantified. I think it could be an important contribution of this paper to estimate the number of deleterious variants that increase in frequency. You have hints of the effect in Fig 3.

In a similar instance, in Table 1, you have novel common SNPs in isolate broken into two groups ** = cases where the none of the general populations have $MAF > 5\%$, and * = cases where the nearest general population has $MAF < 5\%$. This seems like it might be misleading, since most of these SNPs, particularly the latter could have $MAF \sim 4.9\%$. It would be much more interesting to identify the number of variants that are $\geq 5\%$ in the isolate, but rare ($MAF < 1\%$) in all or the nearest general population. The authors refer to the numbers from this table in the text and highlight the as "useful markers for whole-genome association studies in these populations and could potentially lead to novel association signals". I would strongly encourage the authors tone down this statement, because variants that have $MAF > 1\%$ are likely well powered in GWAS in the general population given current sample sizes. It would be especially important if the authors found that a higher proportion of predicted deleterious variants (e.g. by CADD) are common in the isolates (and rare in general populations) compared to neutral variants.

The authors do PCA on rare variants, which highlight recent population structure better than common variants, and should cite O'Connor et al [PMC4327153].

Supplementary Figure 2 (PC 8 vs 7) is great, you don't often see human population structure that looks like the cartoon of a human.

I don't find figure S11 to be particularly helpful. It would be much more helpful to look at the 3 choose 2 pairwise plots. It is very hard to tell what the curvature looks like in your 3D figure.

The authors argue that "Isx is the best overall predictor of the other pairwise measures". They cite Fig 2c, which shows a correlation plot. By eye, it seems like F is actually shifted toward higher correlation than the Isx, but it is hard to tell from a heat map (the constraints of a heatmap). The authors also refer to supplementary figures 12-17, but figures S12-16 seem completely unrelated, and S17 does not compare slow/goodness of fit with other statistics (just Isx). The authors should show that Isx does a better job predicting the other statistics in order to make this claim, because from the data presented I am not convinced.

REVIEWERS' COMMENTS:

Reviewer #1 (Remarks to the Author):

Authors adequately addressed my concerns. I also think they adequately addressed the comments by the other reviewer.

Response to reviewers

Reviewers' comments are shown like this.

Author responses are inset below each comment.

Reviewer #1 (Remarks to the Author):

In their study, Xue et al. analyzed germline variants in 10 isolated populations in comparison with variants in the general population from the 1000 Genomes and UK10K projects. Well executed and described in great detail, albeit mostly in the supplementary material, this large study can serve as a highly useful resource that will no doubt be applied in many future studies.

We appreciate these positive comments.

At the same time, the results presented are incremental, which poses a major weakness of providing too few biological insights. The authors introduce two new measures— DV_{xy} and SV_{xy} —to quantify the diminishing purifying selection in isolates, as well as an additional measure for isolatedness (i.e., Isx), which they suggest can “guide population choice in future complex trait association studies.” Although a potentially important finding, the authors failed to demonstrate how these measures can guide population selection. In response, can the authors provide an example of how using Isx can dramatically improve association studies that use isolated populations instead of the general population?

We agree that this would be a great thing to do, but in order to do this we would need whole-genome sequencing data from thousands samples from multiple isolates and their closest general populations, with the same phenotypic data. There are no such data available at the moment and generating them would be beyond the scope of the current study. However, we have investigated to what extent that it is possible to address this question using available GWAS data: either SNP genotype or whole genome sequence data, or combined. We compiled a list of published GWAS studies on isolated populations (Table below), but found that there is not any suitable dataset among these studies that we could use to address this question. They have different sample sizes (which will lead to different power), different phenotypes, different measures for even the same phenotype, and different data types (different SNP chip, different coverage of whole genome sequencing data etc.). However, most of the GWAS hits identified in the isolates are very rare in the general population, and some of them were only found in the isolates.

We have emphasized the need for such studies in the future by changing our

concluding sentence from “We conclude that the *Isx* value is therefore potentially a reliable indicator of the potential power of association study in the different isolates.” to “We conclude that the *Isx* value is therefore potentially a reliable indicator of the potential power of association study in the different isolates which could be further tested when we have suitable datasets.”

Isolates (ref)	Data	Sample size	Phenotypes	Results	MAF (isolates/general)
Lancaster Amish (ref 4)	Affymetrix 500K Array	809	fasting and postprandial triglycerides	rs76353203	2.4%/0.0%
Icelanders (ref5)	WGS +imputed Illumina SNP chip	4,537 (cases) 54,444 (Controls)	prostate cancer	rs188140481	3.2%/1.0%
MANOLIS (ref 6)	Illumina HumanExome	1,256	HDL levels	rs76353203	1.9%/0.0%
Finnish (ref7)	Illumina HumanExome	8,229	insulin processing and secretion	rs61741902 rs35233100 rs150781447 rs3824420 rs35658696 rs36046591 rs2650000 rs505922 rs60980157	1.4%/1.0%/1.0% 3.7%/5.0%/5.8% 2.0%/1.0%/0.4% 2.9%/0.3%/0.0% 5.3%/4.0%/3.2% 5.3%/4.0%/3.0% 45.5%/36.0%/35.0% 47.1%/37.0%/34.6% 30.0%/24.0%/24.4%
Finnish (ref9)	83 LoF SNPs	36,262	60 quantitative traits	Chr14:55890937 Chr8:17726470 Chr16:84495318 Chr13:100518634 Chr11:59863030	1.1%/0.0% 3.7%/0.0% 2.3%/0.5% 3.5%/2.2% 1.9%/0.1%
Greenland (ref10)	Illumina Cardio-MetaboChip + exomseq	2,575	T2Drelated quantitative traits	rs61736969	17.0%/0.0%

Sardinian (ref11)	4x WGS	2,120	circulating	rs11549407	4.8%/0.0%
			lipid levels	Chr11:116661101	2.5%/0.0%
			and five	rs73198138	0.4%/2.3%
			inflammatory	rs183233091	1.0%/2.3%
			biomarkers	Chr12:125406340	0.7%/0.0%
				rs34599082	3.7%/1.0%
			rs200491743	0.5%/0.0%	
Icelanders (ref12)	WGS +imputed Illumina SNP chip	11,114 (case) 267,140 (controls)	Type 2	rs76895963	1.47%/2.4%
			diabetes	rs75615236	7.10%/7.2%
				rs35658696	4.98%/4.0%
				rs78408340	0.65%/0.0%
				Chr13: 27396636	0.20%/0.0%

Minor comments

1. It is unclear to me which samples were included in the total count of 9,375 from the current study and the 1000G and UK10K projects.

We have made this clearer in the text and updated the Supplementary Table. The total count of 9375 samples includes 3059 from the current study, 2353 from the 1000 Genomes Project Phase 3 release, and 3781 from UK10K. We have added these numbers into the main text.

2. Although I might have overlooked such an explanation, it appears as though the term N_e is not defined in the text.

We added this in the text: it is effective population size (N_e).

Reviewer #2 (Remarks to the Author):

This is an interesting study of whole genome sequencing of multiple isolated European populations along with matched general population controls. The idea was to perform low-coverage sequencing (4-10X) of ~3000 individuals representing a variety of populations, reconstruct their demographic history, and quantify the extent of population-specificity of variants in each isolated population. The authors develop several new statistics to describe the isolatedness and population specificity of variants to discuss the potential relaxation of selective constraint on population isolates. Importantly, the data are made available through EGA. I very much like seeing the demographic inference of these populations, and think the data are summarized well to have broad impact.

We appreciate these positive comments.

Of course there are usually several ways that the study could be improved, and I've tried to identify them below.

First, I don't think the title of the manuscript reflects the results of the paper. The only aspect of "functional" variants in this paper appears to be the use of CADD scores. This is of course a functional prediction, not a true aspect of "function". Further, it is not clear why CADD score >20 is used as a threshold. Lastly, it is not clear that there are any tests of "enrichment" statistically backing up the title's claim.

We agree that the CADD score is not an empirical measure of function, but instead a prediction of function – albeit perhaps the best current option for evaluating the functional importance of variants genomewide, as there is no experimental way to determine the function of variants on a genomewide scale at the moment. Although CADD provides only predictions, it is difficult to doubt that some CADD classes, such as missense and loss-of-function variants in protein coding genes, generally have more functional impact than variants in unannotated non-coding regions.

We fully agree that using a cutoff is always somehow arbitrary, but here we need to both focus on the variants most likely to be functional and roughly balance the number of variants in each bin, to provide compatible statistical power across bins. A score of 20 provides the best compromise. We have added an exploration of other cutoff values, 15 and 25, as well as different bins and find very similar results. We have added three figures to demonstrate this (Supplementary Figure 18)

Besides using CADD scores, we also tested the enrichment using missense variants (DVxy-missense) and missense plus loss of function variants (coding functional variants, DVxy-coding); we found that the both DVxy-missense and DVxy-coding are significantly greater than 1 in almost all of the isolated populations (except GRM, due to the different variant calling used in the GRG; Figure 3C and Supplementary Table 12), which suggested the enrichment of functional drifted variants. We have corrected DVxy statistics using intergenic variants for genetic drift, as can be seen from the formula we used for DVxy calculation:

$$DVxy_coding = \frac{\%DVx\ missense}{\%DVx\ intergenic} \bigg/ \frac{\%DVy\ missense}{\%DVy\ intergenic}$$

We used the jack-knife bootstrap approach to get 95th percentiles of all these estimates. If the values are greater than, and do not overlap with, 1, they are statistically significant. These estimates for functional variants (either CADD scores greater than 20, or missense, or missense plus LoF) are all significantly greater than 1 in the isolates, but smaller than 1 in the general populations. We have made it clearer in the main text that such a test is used.

We see these findings as strong evidence for enrichment of functional variants at least for the subset we are particularly interested in here, the DVs with 2-5% frequency in isolates, yet 3 times higher than in its general population. To return to the reviewer's initial point about the title, we thus think that the original title is justified, and note that the predictions do not need to be 100% accurate for it to be appropriate. However, we don't see this as a key point and could change it to "Choosing the best populations for association studies – insights from large-scale whole-genome sequencing of multiple isolated European populations", which is more neutral but longer and less catchy. We would welcome editorial input on this.

Second, I would argue that the authors should be more precise with their use of terminology when discussing relaxation of purifying selection. One measure of natural selection is the fitness effect of a given mutation (s). This is not something that is observable, and is actually quite hard to infer. The "strength" of selection is often thought of in terms of the product of the fitness effect and the effective population size ($Ne*s$). For a population isolate, the extreme bottleneck and subsequent inbreeding can result in a sustained reduction in Ne , which of course reduces the product of $Ne*s$, but such a bottleneck or any other demographic effect is unlikely to impact s .

We fully agree that drift is a confounding factor in a study of purifying selection, especially in isolated populations, so when we developed the DVxy and SVxy statistics, we took this into account and corrected for the genetic drift effect. When we calculated DVxy for missense variants, or missense plus loss of function variants, in each isolate or general population, we used intergenic variants to correct for the effect of genetic drift, which affects all variant equally. When we calculated DVxy for the whole genome using different CADD scores, in each isolate and general population, we saw differences of the DVxy value between each isolate and its general population increase significantly from low to high CADD scores. This itself is a correction: if there is no selection on the functionally important variant (more likely with higher CADD scores), we should not see any difference between isolate and general population. When we calculated SVxy, we used synonymous variants to correct for genetic drift in a similar way.

We also agree that these corrections and the functional annotation are not perfect, but nevertheless, the overall trend is very striking.

The authors find that, consistent with Do et al [29], there is no evidence for reduction of genetic load across individuals. It therefore seems incorrect to conclude (as per the abstract) that the authors "demonstrate the relaxation of purifying selection". In fact, what the authors are seeing is that some deleterious variants increase in frequency due to drift, but the small population size results in a reduction in the introduction of new deleterious alleles (thereby having no effect on genetic load). This is a result that is also

similar to Simons et al. [PMC3953611].

If we consider all variants in each population, and ask whether we see an increased genetic load per individual, we indeed do not see any such increase which is consistent with Do et al. [29] as the reviewer points out. However, when we only look at a particular subset of the variants (frequency 2-5% in the isolated population, yet three time higher than in its general population), called drifted variants (DVs) here, we see the enrichment of functional DVs in the isolates. This is due to the site frequency spectrum in the isolate and its general population being different.

Major comments:

On page 4, the authors argue that IF1-IF4 exhibit a “greater level of isolation and lower level of gene flow with their general population”. However, IF1-IF4 have the smallest sample size (by a factor of >6 on average). I am worried that comparisons like this and other places that differences in sample size could be playing a major role in population differences. The general populations that the authors sample are far larger than any of the isolates, and so it is unclear to what extent the results are driven by sample size. For some of the analyses, I think it would be very helpful to have a supplemental figure showing what happens to figures 1d, 2b, 3c, and 3d if all populations were down sampled to 36 individuals (the size of IF4, your smallest sample size in Table 1).

This is potentially a serious concern, and the one that we took into account in our study design. Of the specific examples the reviewer highlights, Figure 1d and 3d are both based on sample sizes of 36 from each population, but Figures 2b and 3c were not. We have added Supplementary Figure 18a and Supplementary Figure 20a for 2b and 3c. We have also redone the DV_{xy}-wg with different CADD score bins using the minimum sample size (Supplementary Figure 18b). As the reviewer suggests, the overall results are pretty similar.

Around lines 166, the authors say “several lines of evidence suggest relaxed purifying selection in the isolates due to their reduced N_e , although as expected we do not detect substantially increased genetic load per genome...”. It seems that these are directly contradictory statements. I don’t see how you can have relaxed purifying selection without increased load. Instead, it seems like you have equivalent levels of purifying selection, but increased drift due to lower N_e . It would be great to see how the authors differentiate relaxed purifying selection from a lower N_e that causes increased drift (so most deleterious variants are lost, but some increase in frequency) and a decreased introduction of novel deleterious variants. The statistics proposed do not seem distinguish these two, and seem consistent with my alternative explanation. I would argue that your results are highly interesting and valuable either way, but one just needs to be careful with terminology to be consistent with recent literature unless you

have strong evidence refuting recent literature (and I do not see that being the case here).

This is a good point and we should make our analyses clearer. The genetic load was measured at the per-individual/genome level using all variants, and we did not find any increase, as expected. Genetic drift and relaxation of purifying selection in the isolated populations change the frequency spectrum of deleterious variants at the population level: some deleterious variants increased their frequency and others decreased or were lost, so did not significantly change the number of deleterious variants per individual.

As the smallest sample size among our populations is 36, and in order to avoid the bias from different sample sizes, we subsampled 36 individuals from all of the populations studied here in several of the analyses. This limited our power to test for relaxation of purifying selection among all variants, so we concentrated on subsets which could give us the best power. One such subset is the DV class, which are at frequency 2-5% in the isolated population, yet three time higher than in its general population. Here, we found significant enrichment for functional variants (missense, missense plus loss-of-function) in the isolates. The other statistic we used for testing relaxation of purifying selection is SV_{xy} , where we only look at singleton variants in coding regions. Both statistics we have corrected for the drift effect either by intergenic variants, or synonymous singleton variants as described in the comments above.

The authors suggest that they do not find convincing evidence for positive selection in any isolate using two statistics. While this is consistent with some arguments in the literature (e.g. Hernandez et al [PMC3669691]), it would also be helpful to try a couple other statistics, particularly the SDS metric from Field et al [PMID: 27738015] recently presented (likely published after submission of this paper, though the preprint and method have been available for a while).

We agree with the reviewer that the SDS method is a powerful way to detect very recent selection and soft positive selection on standing variants; however, it requires very large sample sizes, which we do not have for our isolates. But we still tried running this method on one of our isolates (UKO) and one general population (UKG, which is the UK10K data, the same as the SDS paper used). We successfully replicated the findings in the lactose tolerance SNPs in the UKG (UK10K), but failed to find any convincing signal in the UKO, although we do expect lactose tolerance to be selected in this population as well since Orcadians enjoy dairy products (e.g. <http://www.orkneyfoodanddrink.com/orkney-creamery>). The derived allele frequency of this SNP is very similar in UKG and UKO (0.73 and 0.78, respectively). UKO is the isolate for which we have the largest sample size (397) and second lowest isolation index (genetic drift effects should be smaller), so SDS should have the best power among all of the isolates

we studied. This negative finding indicates that SDS is unlikely to have enough power to detect signals in the even smaller samples. But we added a section to the Supplementary Material with these results and some discussion (section 5.2.3), and a sentence to the main text.

Minor comments:

I am not a fan of approximate numbers in an abstract, and think people would appreciate seeing “Here, we perform a comprehensive investigation using low-depth whole genome sequencing of 1433 individuals from eight European isolates and 1663 individuals from two matched general populations.” You then integrate data from a couple other population isolates and 1000 Genomes Project data, and I’m not sure how best to indicate that here.

We appreciate this suggestion and avoided using approximate numbers in our abstract, but the word limit restricts what we can do. So we changed the sentence to “Here, we perform a comprehensive investigation using 3059 newly-generated whole-genome low-depth sequences from eight European isolates and two matched general populations, together with published data from the 1000 Genomes Project and UK10K.”

In the first paragraph, the authors say “In isolated populations (isolates), some deleterious rare variants have increased in frequency while some neutral rare variation is lost...”. The first part is likely to be a minority of cases, but I don’t see this quantified. I think it could be an important contribution of this paper to estimate the number of deleterious variants that increase in frequency. You have hints of the effect in Fig 3.

In its context, this is not a conclusion we are drawing, but rather a prediction. Isolated populations have experienced a bottleneck, which will increase genetic drift, and by chance some deleterious rare variants will have increased in frequency while some neutral rare variation is lost. However, we appreciate the interest in this topic and have generated some relevant numbers. We defined missense plus LoF variants as functional (most would be deleterious) for the coding region, and CADD scores greater than 15 for genomewide variants. We found that around ~150-~700 deleterious variants in coding region and ~500-~2800 genome wide have drifted to higher frequency in the isolates (these variants have MAF >5.6% in isolates, but <1.4% in all of the four general populations). We added this table to the Supplementry Material (Supplementary Table 4), and added a sentence to the main text.

Isolates	Total	Missense plus LoF	CADD score 15
FIK	70579	410 (0.58%)	1479 (2.1%)

GRM	49884	266 (0.53%)	988 (2.0%)
IF1	119157	689 (0.58%)	2676 (2.2%)
IF2	94496	518 (0.55%)	2080 (2.2%)
IF3	107281	616 (0.57%)	2417 (2.3%)
IF4	122254	688 (0.56%)	2792 (2.3%)
IVB	30284	154 (0.51%)	530 (1.8%)
UKO	36512	210 (0.58%)	634 (1.7%)

In a similar instance, in Table 1, you have novel common SNPs in isolate broken into two groups ** = cases where the none of the general populations have MAF>5%, and * = cases where the nearest general population has MAF < 5%. This seems like it might be misleading, since most of these SNPs, particularly the latter could have MAF ~4.9%. It would be much more interesting to identify the number of variants that are >=5% in the isolate, but rare (MAF<1%) in all or the nearest general population.

This is a good suggestion. In order to avoid sample size bias, we did these analyses using the minimum sample size set, so we only have 36 individuals from each population. We have now redone the analyses by counting the variants which have alternative allele count of 4 or more in the general population (equivalent to MAF > 5.6%), but 1 or less in the general population (equivalent to MAF <1.4%) and updated Table 1 accordingly. The numbers are lower than before, but still quite substantial.

The authors refer to the numbers from this table in the text and highlight the as “useful markers for whole-genome association studies in these populations and could potentially lead to novel association signals”. I would strongly encourage the authors tone down this statement, because variants that have MAF>1% are likely well powered in GWAS in the general population given current sample sizes.

We agree with this point, so we toned down this statement and changed it to “useful markers for whole-genome association studies in these populations and some of them (if absent from the general population) could potentially lead to novel association signals.”

It would be especially important if the authors found that a higher proportion of predicted deleterious variants (e.g. by CADD) are common in the isolates (and rare in general populations) compared to neutral variants.

We have done this, but in a slightly different way. We looked at variants at 2-5% in the isolates, yet at least 3 times higher than in its general population (Figure 3b). We seen the enrichment in the variants with higher CADD scores (more likely functional important or deleterious), but we did not see the difference for the ones with low CADD scores (more likely neutral). We appreciate that we have only

done this with subset of the variants; however, the variants we use are the ones that give us the best power to address the question, given the sample sizes we have.

The authors do PCA on rare variants, which highlight recent population structure better than common variants, and should cite O'Connor et al [PMC4327153].

Thanks for pointing this out; we have added this reference.

Supplementary Figure 2 (PC 8 vs 7) is great, you don't often see human population structure that looks like the cartoon of a human.

Thanks for such a nice comment.

I don't find figure S11 to be particularly helpful. It would be much more helpful to look at the 3 choose 2 pairwise plots. It is very hard to tell what the curvature looks like in your 3D figure.

We have generated a new figure for this, with three panels each showing a pairwise plot. It is much clearer and we appreciate this suggestion.

The authors argue that "Isx is the best overall predictor of the other pairwise measures". They cite Fig 2c, which shows a correlation plot. By eye, it seems like F is actually shifted toward higher correlation than the Isx, but it is hard to tell from a heat map (the constraints of a heatmap).

We have added the correlation coefficient values into the heat map and to Supplementary Table 11 to make the information clearer.

The authors also refer to supplementary figures 12-17, but figures S12-16 seem completely unrelated, and S17 does not compare slow/goodness of fit with other statistics (just Isx).

Thank you for pointing this out: the references are indeed wrong. We have corrected these and also added Supplementary Table 11 to show the pairwise correlation coefficients.

The authors should show that Isx does a better job predicting the other statistics in order to make this claim, because from the data presented I am not convinced.

We agree with the reviewer that Isx does not really stand out strikingly in our current study, showing similar prediction power to the inbreeding coefficient (F) and runs of homozygosity (ROH) overall. Nevertheless, Isx was calculated from

three population demographic parameters, which cover different aspects of the relevant population features. It should thus be more robust to population confounding factors. In contrast, both of F and ROH are only based on a single measurement, and thus more prone to confounding factors such as recent inbreeding, or consanguinity. The populations and samples in the current study don't have such confounding factors, so Isx does not stand out as much as in might. We have therefore toned down this claim and suggest that it should be further tested in future studies.

REVIEWERS' COMMENTS:

Reviewer #1 (Remarks to the Author):

Authors adequately addressed my concerns. I also think they adequately addressed the comments by the other reviewer.

We thank to the reviewer for these positive comments